# Emergent and Neglected Equine Filariosis in Egypt: Species Diversity and Host Immune Response

**DOI:** 10.3390/pathogens11090979

**Published:** 2022-08-27

**Authors:** Faten A. M. Abo-Aziza, Seham H. M. Hendawy, Hend H. A. M. Abdullah, Amira El Namaky, Younes Laidoudi, Oleg Mediannikov

**Affiliations:** 1Department of Parasitology and Animal Diseases, Veterinary Research Institute, National Research Centre, Dokki, Giza 12622, Egypt; 2Tick and Tick-Borne Diseases Research Unit, Veterinary Research Institute, National Research Centre, Dokki, Giza 12622, Egypt; 3Aix Marseille Université, IRD, AP-HM, MEPHI, IHU-Méditerranée Infection, 13005 Marseille, France; 4PADESCA Laboratory, Veterinary Science Institute, University of Constantine 1, El-Khroub 25100, Algeria

**Keywords:** equids, zoonotic filariosis, cytokines, phylogeny, *Mansonella* sp., *Setaria digitata*, *Dirofilaria repens*

## Abstract

Equine filariosis (EF) is a neglected vector-borne disease caused by nematode species belonging to the Onchocercidae and Setariidae families. Aside from their zoonotic potential, some species are responsible for serious health problems in equids worldwide, leading to significant economic difficulties. Here, we molecularly investigated equine blood samples (320 horses and 109 donkeys from Egypt) and four adult worms isolated from the peritoneal cavity of 5 out of the 94 slaughtered donkeys. In addition, quantitative enzyme-linked immunoassays (ELISAs) targeting circulating cytokines were used to identify whether the immunological profile of the infected animals is a Th1 (i.e., INF-gamma as indicator) or Th2 (i.e., IL-5 and IL-10 as indicators) response type. Overall, 13.8% and 0.3% of the donkeys and horses, respectively, were scored as positive for filaroid DNA. The 18S phylogeny revealed the occurrence of three different filaroid species, identified here as *Mansonella* (*Tetrapetalonema*) sp., *Setaria digitata* and *Dirofilaria repens*. Th1 (INF-gamma and IL-5) and Th2 (IL-10) immune response types were identified in equines infected with *S. digitata* and *Mansonella* (*T.*) sp., respectively. These results provide new data on the species diversity of EF in Egypt and extend knowledge of the downregulation of the protective immune response by the potentially zoonotic *Mansonella (T)* sp. There is an urgent need to implement control measures to preserve equine health and limit the propagation of these vector-borne filaroids in Egypt.

## 1. Introduction

Equine filariosis (EF) is a neglected vector-borne disease caused by heteroxenous parasitic nematodes belonging to the Onchocercidae and Setariidae families. Adult nematode parasitise tissues and the body cavities of equids produce skin- or blood-dwelling microfilariae. The larvae are ingested by blood-feeding arthropods and undergo two-stage development followed by the third stage (L3), where they are transmitted to a new receptive host [1,2,3,4,5]. The most known filarioid species infecting equids belong to the *Onchocerca*, *Dirofilaria* and *Setaria* genera. Infection by these parasites leads to a myriad of problems ranging from fibrinous myocarditis and/or peritonitis to tendinitis when the worms infect the muscles and ligaments [6]. In all cases, the infection is often responsible for considerably limited movement and lameness [7,8,9]. In addition, the unusual localisations of the worms in the eye and the central nervous system may occur and may lead to blindness and neurological disorders [10,11]. Moreover, the deep localisation of the adult worms inside the body cavities and tissues and the heterogeneous localisation of their microfilariae require invasive methods, such as skin biopsies or the surgical removal of the parasite, which represents a major limitation for the routine diagnosis of such infection. Therefore, in the absence of specific serological tests, the use of molecular identification remains the most effective and accurate method for identifying these parasites from blood or tissue fragments of infected animals [12,13,14]. This significant lack of knowledge of their epidemiology and biology makes their control and management difficult.

Host–parasite interaction relies on the expressed cytokine profile which is modulated by the parasite itself [15,16,17]. The predominant immunological response during filarial infection is antigen-specific Th2 stimulation, leading to an expansion of IL-10, along with the inhibition of Th1 response, Babu and Nutman [18]. This explains the parasite longevity within the infected hosts [19,20]. It is admitted that filarial parasites produce and secrete a range of immunomodulatory molecules, such as the ES-62, that interfere directly with the natural immune response by suppressing the inflammatory phenomenon and countering the pathology associated with the disease [15]. However, host–parasite interaction differs between species, resulting in the specific modulation of the immune response according to the filaroid species. For example, the production of the cystatins by some filarial parasites such as *Onchocerca volvulus* and *Acanthocheilonema viteae* leads to the inhibition of the cysteine protease group (i.e., legumains and cathepsin L and S) and induces the production of anti-inflammatory interleukin (i.e., IL-10) [21,22]. The precise immune mechanisms that govern *S. equina* infection in donkeys have been partially investigated and depend on the TNF-α and IL-4 as Th1/Th2 balance with regard to the existence of the adult stages [6]. For example, *S. equina* asymptomatic infections are often characterised by the down- and upregulation of TNF-α, and IL-4, respectively [6], while chronic infections are associated with an increase in IL-10 expression, leading to the down- and upregulation of IFN- γ (Th1) and IL- 5 (Th2), respectively [15,23].

In Egypt, equids play a significant role in numerous sectors, including agriculture, the police services, tourism and the pharmaceutical and transport sectors, as well as in recreational activities (i.e., sport, gaming and entertainment) [24,25]. Thus, understanding the impact of infectious pathogens on equines should not be overlooked as a vital part of Egypt’s culture and economy. Studies on neglected filaroid parasites (i.e., *Setaria equina*, *Onchocerca cervicalis*, *Onchocerca reticulata* and *Parafilaria multipapillosa*) remain scant and outdated from many Egyptian provinces [7,12,26,27,28,29,30,31,32]. Therefore, this study aimed to investigate the species diversity of equine filariod and their impact on the host immune response of both horses and donkeys from Egypt.

## 2. Results

### 2.1. Prevalence and Phylogeny of Filaroids

Overall, 22 (5.1%, 95% CI) (1 horse and 21 donkeys) out of 429 equids were investigated and tested positive using pan-filarial qPCR. Filaroid DNA was successfully amplified from 16 out of the 22 qPCR-positive blood samples, and all (*n* = 4) retrieved worms using the 18S primers. Infection rates of 3.7% (0.3% in horse (1/320) and 13.8% in donkeys (15/109)) were recorded. Likewise, all DNA amplicons were sequenced and grouped into three sequence groups according to BLAST analysis (Table 1): group i) amplified from five donkeys and one worm retrieved from a donkey’s peritoneal cavity and showed 100% (739/739) and 99% (736/737) identity with *S. digitata* from the UK (GenBank DQ094175) and Egypt (GenBank MN728217), respectively. Group ii) amplified from a microfilaremic donkey and a worm from a donkey’s peritoneal cavity and were 100% (739/739) identical with those of *D. repens* previously detected in dog from Egypt (GenBank MN728215). Group iii) amplified from 10 blood samples (one horse and nine donkeys) and one worm retrieved from a donkey’s peritoneal cavity and were 100% (1057/1057) identical to *Mansonella* sp. (genotype OM- 2015, GenBank MT786947-49), which is detected in donkeys from Algeria (Table 1). Accordingly, the maximum likelihood phylogeny confirmed the BLAST results for the three filaroid species (*Mansonella* sp., *S. digitata* and *D. repens*) with high bootstrap values (Figure 1). In addition, the ML phylogeny resolved the species identification of the *Mansonella* sp. herein isolated at the subgenus *Tetrapetalonema* (Figure 1).

### 2.2. Immunological Studies

The slopes of regression (b) among the graded log-doses and their corresponding binding were 0.467, 0.529 and 0.402 for INF-gamma, IL-5 and IL-10, respectively. *Setaria digitata* infection induced a higher inflammatory response, as INF-gamma was significantly higher in *S. digitata*-infected equines (*p* < 0.05) compared to controls. Both *Mansonella* (*T*) sp. and *S. digitata* infection resulted in an increase in Th2 cytokines, but the elevation in *S. digitata*-infected animals was more pronounced. It was found that the IL-5 level in *S. digitata* was significantly higher (*p* < 0.01) than in the control group. Similarly, equines infected with *Mansonella* (*T*) sp. showed a significant elevation in the IL-5 level relative to the control group (*p* < 0.05). It was observed that equines infected with *Mansonella* (*T*) sp. showed a significant elevation in IL-10 level (*p* < 0.01) compared to the control group (Table 2). Moreover, comparing the effect of the infection with the two filarial species in terms of cytokine levels, it was found that INF-gamma and IL-5 levels were significantly higher in equines infected with *S. digitata* than those of equines infected with *Mansonella* (*T*) sp. (*p* < 0.05). However, *Mansonella* (*T*) sp. infections resulted in a significant elevation in IL-10 level (*p* < 0.01) compared to *S. digitata* infections. In addition, Th1/Th2 in *S. digitata* was 1.99, while in *Mansonella* (*T*) sp., Th1/Th2 was 1.71 (Table 2). In the control group, a moderate positive correlation was observed between INF-gamma and IL-5, though a weak correlation between INF-gamma and IL-10 serum levels was recorded. A strong positive correlation was recorded between INF-gamma, IL-5 and IL-10 serum levels in animals infected with *S. digitata*. In contrast, a moderate positive correlation between INF-gamma and IL-5 serum levels and a negative moderate correlation between INF-gamma and IL-10 serum levels were observed in *Mansonella* sp. (*T*)-infected groups (Figure 2).

## 3. Discussion

Equine filariosis is a neglected vector-borne disease causing a reduction in the working capacity of equids. Consequently, health threat and economic losses are the main observed problems in endemic areas [32,33]. Therefore, their accurate detection coupled with a better understanding of cytokine profiles and the immunological signalling associated with the disease should not be overlooked for disease management and control [6,34]. The present study provides new data on the species diversity and disease-associated cytokines of EF from Egypt.

In the current study, the overall prevalence of EF was 3.7%, with a higher prevalence among donkeys (13.8%) than among horses (0.3%). Based on our designed 18S rRNA primers, three distinct filaroid species were detected in equines: *Mansonella* sp., *S. digitata* and *D. repens*.

The infection rate of *Mansonella* was 0.3% (1/320) in horses from Cairo and 8.3% (9/109) in donkeys from Beni Suef and Al-Faiyum. Phylogenetically, this species clustered together with *Mansonella* sp. OM-2015 genotype and belongs to the *Tetrapetalonema* subgenus within *Mansonella* (*T*.) *atelensis* (GenBank: MT336173). According to GenBank entries, the *Mansonella* sp. OM-2015 genotype was identified either in donkeys from Algeria using 18S, ITS1, *cox1* and 5S sequencing (genotype OM-2015, MT786947-49; MT786950-51, 55–58; MT791217 and MT795713, 15–17, respectively), and from Senegal using ITS1 sequencing (genotype OM-2015, MT786952), in horses from Senegal using ITS1 and 5S sequencing (genotype OM-2015, MT786953-54 and MT795714, respectively) or in biting midges *Culicoid esenderleini* from Senegal using generic ITS1 sequencing (genotype OM-2015, KR080175). Despite the scant data, this species seems to be widely distributed and was only identified in equids (i.e., donkeys and horses) and in biting midges *C. enderleini*, suggesting the reservoir and vector role of these hosts, respectively (Figure 3).

This phylogenetic analysis resolved the identification of the *Mansonella* sp. genotype OM-2015 at the subgenus level (*Tetrapetalonema*). However, in the absence of morphological data on this species isolated in this study and given the absence of DNA sequences from the other morphemically described *Mansonella* (*T*) species, the possibility of the *Mansonella* (*T*) sp. described in this study being a new species cannot yet be ruled out. The subgenus *Tetrapetalonema* encompass 14 valid species all isolated from South American monkeys, with the exception of *Mansonella* (*T*) zakii [35,36] (syns. *Parlitomosazakii* [35]; *Tetrapetalonema zakii* [35,37]; *Dipetalonema zakii* [35]) in *Leontopithecus* (= *Leontocebus*) *rosalia* (Linnaeus), which was isolated in Egypt (in captivity, originating from Brazil) [35] and is considered to be a species studied by Eberhard and Orihel [36]. More studies are needed to confirm or describe this potential new species of the *Mansonella* group and determine its importance in pathology.

We detected *S. digitata* and its microfilariae in donkeys derived from Al-Faiyum province, with a total prevalence rate of 4.6% (5/109). Setariosis caused by different *Setaria* spp. is a filarial disease affecting bovines and can accidentally be transmitted into unusual hosts such as horses, donkeys, sheep, goats, and camels [38,39,40,41]. *Setaria digitata* has been detected in horses in Korea [42,43], Malaysia [8], India [13], Iran [5], and China [9]. *S. equina* has been detected in donkeys in Egypt [7,12,32]. The identification of adult *S. digitata* in donkeys and microfilariae in the blood indicate that donkeys are competent hosts of *S. digitata* in Egypt. The relatively high prevalence of *S. digitata* microfilaraemia in donkeys in Egypt (4.6%) may lead to a significant impact on animal health.

We detected adult *D. repens* and its microfilariae in one donkey from Al-Faiyum province, with a total prevalence rate of 0.9% (1/109). To the best of our knowledge, *D. repens* has never been reported in donkeys anywhere in the world. *D. repens* affects dogs and other carnivores, causing subcutaneous dirofilariasis [44,45]. Other mammals, including humans, may also be affected [46], but they very rarely become the final hosts or develop microfilaraemia [47]. A generalisation stating that donkeys play a significant role as a reservoir of subcutaneous dirofilariasis cannot be made from the single animal we describe here; however, it is evidence that this animal may also be a competent host of *D. repens*.

An assay of cytokine responses (INF-gamma as Th1 and IL-5 and IL-10 as Th2 indicators) to *Mansonella* (*T*) sp. and *S. digitata* infection was performed to elucidate immunological differences, motivating an immune response. In this study, it was noted that equines infected with *Mansonella* (*T*) sp. showed a significant elevation in IL-10 levels compared to the control group. Moreover, comparing the effect of the infection with the two filariod species on cytokine levels, it was found that INF-gamma and IL-5 levels were significantly higher in equines infected with *S. digitata* than those infected with *Mansonella* (*T*) sp. It is well known that polymorphous cells are essential for killing adult worms through the evoked inflammatory nodule formation and production of IL-5 in response to helminths [48]. IL-5 is also essential for containing parasitaemia [49]. In IL-5-deficient mice, microfilaraemia was greatly enhanced and pronounced in addition to increased adult worm burden [50,51]. In contrast, many studies have shown that Th 1 cytokines (INF-gamma) can be involved in the protection against helminths through proper neutrophil migration [52]. However, *Mansonella* (*T*) sp. infection was correlated to a significant elevation of IL-10 level compared to *S. digitata* infection. In addition, Th1/Th2 in *S. digitata* was 1.99, but in *Mansonella* (*T*) sp., Th1/Th2 was 1.71. In the control group, a moderate positive correlation was observed between INF-gamma and IL-5, while a weak correlation between INF-gamma and IL-10 serum levels was recorded. A strong positive correlation was recorded between INF-gamma and IL-5 and IL-10 serum levels in animals infected with *S. digitata*. In contrast, a moderate positive correlation between INF-gamma and IL-5 serum levels and a negative moderate correlation between INF-gamma and IL-10 serum levels were observed in *Mansonella* (*T*) sp. infected groups. Helminths are able to downregulate host immune answers in order to facilitate their long-term existence. This ability to modulate the host immunological situation is a key part of the evolutionary achievement of helminths and could be attributed to immunomodulatory mediators such as ES-62, cystatins, legumains and cathepsin L and S, which are secreted by the nematode parasite as part of their immune-evasion strategy. They have an immunomodulatory effect to enhance the production of anti-inflammatory IL10 [17,22]. Such a strategy was more pronounced in *Mansonella* (*T*) sp. than in *S. digitata*. Inflammatory procedures are blocked and pathogenic damage to the host is reduced or even subclinical. This ultimately lets the parasite continue within the host for a long time [53]. Mechanisms that downregulate the immune responses include the induction of regulatory T cells [54]. T helper cells constitute common T lymphocyte activation followed by differentiation into Th1 and Th2 phenotypes associated with the progress of type-2 cytokines and the impairment of type-1 cytokine production [55]. This shifting plays a key role in regulating the balance between infection and disease. In the case of Th1 and Th2 phenotypes, they are predominantly related to susceptibility and protection, respectively [56]. Subash and Nutman [57] reported that early filarial infection was accompanied by the elevation of Th1 over Th2 cytokines, which is essential to understanding the pathogenesis of infection and the host–parasite relationships. This response could be the beginning of acute filariasis and the formation of host resistance to the helminth infection [57]. Th1 and Th2 cytokines orchestrate different immune pathways to fight Strongylus; Th1 cytokines coordinate cellular immune responses, and Th2 cytokines coordinate humoral immune responses [58]. Helminths infections are usually associated with polarised Th2/Th1 immune responses [16,57]. In this study, Th1/Th2 was 1.99 in *S. digitata* and 1.71 in *Mansonella* (*T*) sp. These results indicated that both infections directed the effort of cytokines to the humoral immune response, as reflected by the elevation of IL-5. Conversely, this effort was reflected in cellular immune response, as indicated by the measured balance.

## 4. Materials and Methods

### 4.1. Study Design and Animal Sampling

Between 2016 and 2018, a cross-sectional study was conducted on equids from three Egyptian provinces (Cairo, Beni-Suef and Al-Faiyum; Figure 4). Using the convenience sampling strategy, a total of 429 equids, including 320 live horses and 109 donkeys (94 donkeys slaughtered at the zoo and 15 live donkeys), were randomly assigned to the study (Table 3). EDTA-blood and sera samples were, respectively, collected either from the jugular vein or the carotid arteries/jugular vein of live equids and the donkeys’ corpses immediately after slaughter. The collected blood and serum samples were stored at −20 °C until analysis. Adult filaroids were searched for in all donkey corpses, and when found were stored in 70% alcohol until further investigation.

### 4.2. Molecular Studies

#### 4.2.1. Polymerase Chain Reactions and Sequencing

First, all EDTA-blood samples (200 µL) and a piece of tissue (~25 mg) from each retrieved worm were subjected individually to mechanical and enzymatic lysis steps prior to DNA extraction as previously described [4]. DNA extraction was performed by the EZ1 biorobot Qiagen (Hilden, Germany), using the EZ1 DNA Tissue Kit Qiagen (Hilden, Germany) according to the manufacturer’s instructions. DNA was eluted in 200 µL and stored at −20 °C until molecular analysis.

All DNA samples were molecularly screened for filaroid DNA using the pan-filarial real time qPCR (Pan-Fil 28S qPCR), targeting the 28S rRNA gene as described by Laidoudi and colleagues [45]. All samples which were filaroid-positive by the Pan-Fil 28S qPCR were subjected to PCR amplification using the newly designed primers (Fwd-18S-Nem.58: AATGGTGAAACCGCGAAC and Rwd-18S-Nem.998: AACACCGCTTGTCCCTCTAA). Briefly, primers were designed according to PCR design protocol described by Laidoudi and colleagues [45] and targeting a partial (1325–35 bps) small subunit (SSU) 18S rRNA gene of nematodes. PCR reactions were carried out in a total volume of 50 µL, consisting of 1 µL of each primer (50 nm), 25 µL of AmpliTaq Gold^®^ 360 Master Mix (Thermo Fisher Scientific, Applied Biosystems, Foster City, CA, USA), 18 µL of DNAse-RNAse free water (Eurogentec, Liège, Belgium) and 5 µL of DNA template. PCR reactions were performed using a thermocycler (Applied Biosystems, Paris, France) with initial denaturation at 95 °C for 15 min, followed by 40 cycles including denaturation at 95 °C for 1 min, annealing at 55 °C for 30 s, and elongation at 72 °C for 1.5 min, followed by a final extension at 72 °C for 10 min. Positive and negative controls were included in each amplification. PCR amplification was confirmed in 1.5% agarose electrophoresis.

Finally, NucleoFast 96 PCR plates (Macherey Nagel, EURL, Hoerdt, France) were used for the purification of PCR products, in accordance with the manufacturer’s recommendations. The purified PCR products were sequenced using the Big Dye Terminator Cycle Sequencing Kit (3130X1 Genetic Analyzer, ABI PRISM) within the ABI automated sequencer (Applied Biosystems). The obtained sequences were assembled and edited using ChromasPro software (ChromasPro 1.7, Technelysium Pty Ltd., Tewantin, Australia). The corrected sequences were subjected to preliminary analysis using the BLAST server [59].

#### 4.2.2. Phylogenetic Analyses

Molecular sequences obtained via 18S PCR sequencing from worms and filaroid-positive samples were aligned against the representative members of the five Onchocercidae clades retrieved from the GenBank database using MAFFT v7.490 [60], with adjustment to the direction of the first sequence. Bioedit software (Bioedit version 7.2.5) was used to manually refine the multisequence alignment [61]. The maximum likelihood phylogeny was performed using IQTREE (IQ-TREE multicore version 1.6.12 for Mac OS X 64-bit) software [62], under 1000 Ultra-Fast bootstrap replications [63]. The K2P + R2 model was selected by ModelFinder before computing the tree [64]. DNA sequences of *Ascaris* sp. (GenBank accession number: JN256985), *Toxascaris leonine* (GenBank accession number: JN256984), *Abbreviata caucasica* (GenBank accession number: MN956825) and *Gongylonema nepalensis* (GenBank accession number: LC278392) were used as out-groups to root the tree. Finally, the tree was annotated within iTOL v5 software [65].

### 4.3. Immunological Study

Commercially available sandwich-type enzyme immunoassay kits using biotin-streptavidin chemistry (Genorise Scientific, INC., Glen Mills, PA, USA) were used to quantify the level of serum INF-gamma, IL-5 and IL-10 according to the manufacturer’s instructions. Optical density was measured at a wavelength of 450 nm. To compare the amount of circulating interleukins, the slope of regression “b” of the log dose response curves (*n* = 5) was obtained from a respective standard preparation, calculated according to the method described by Spiegel [66]. In addition, the correlation of the level of circulating INF-gamma to IL-5 and IL-10 in control and filaroid-positive equids was statistically performed using the Minitap programme version 21. Finally, data from each cytokine concentration were presented as mean ± and standard errors (SE). Descriptive statistics and simple one-way analysis of variance (ANOVA) were performed using the SPSS programme version 19 (IBM Corp., Armonk, NY, USA) to characterise the immunological profile associated with each infection.

## 5. Conclusions

Our study provided an epidemiological picture of species diversity and immune response modulation in equids that were naturally infected with filaroid parasites (*Mansonella, Setaria* and *Dirofilaria*) from Egypt. In addition to *S. digitata*, which is a well-known filaroid infecting equids, the presence of both adult and larval forms of *Mansonella (T)* sp. and *D. repens* extends knowledge on the host suitability and reservoir role of equids for these zoonotic/potentially zoonotic species. In contrast to *S. digitata*, *Mansonella (T)* sp. seems to downregulate the protective immune response. However, the pathogeny of EF in general and equine mansonellosis caused by *Mansonella (T)* sp. remains unclear, and further studies on the biology of these equine vector-borne nematodes are needed. Likewise, further studies are needed to assist in the quest to identify the elusive vector of the *Mansonella (T)* sp. as well as to promote its full morphological description.

## Figures and Tables

**Figure 1 pathogens-11-00979-f001:**
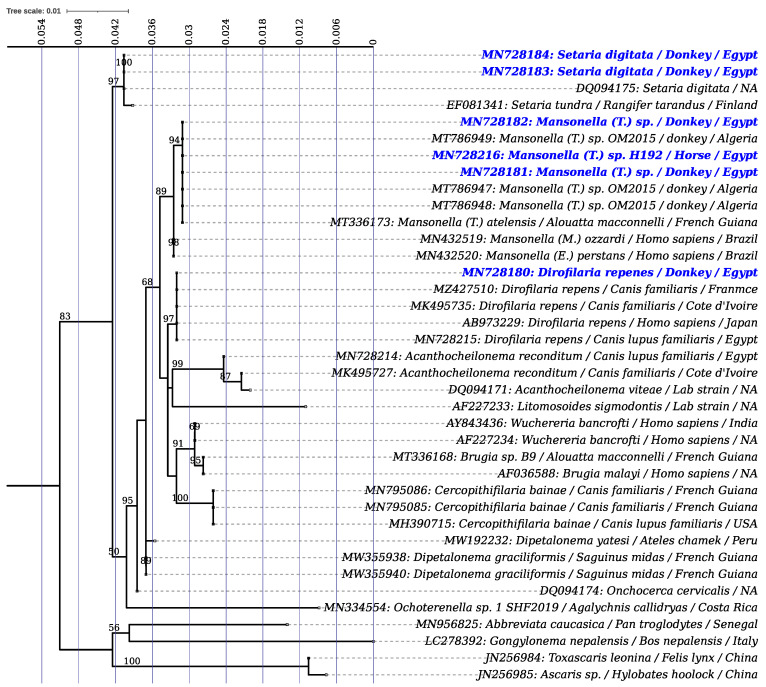
The 18S-rRNA-based phylogeny of filaroid species identified in this study and the representative members of the five Onchocercidae Clades (ONC1-5) available in the GenBank database. The tree corresponds to the IQTREE inferred from 39 partial (713 bps) sequences with 9.7% (i.e., 52 and 71 of parsimony informative and distinct sites, respectively) of informative sites using the K2P (+R2) model under 1000 bootstrap replicates and the ML method. GenBank accession number, species name, host, and geographical origin are indicated at the tip of each branch when available. Blue bolded label indicates sequence types are amplified in the present study. Bootstraps values higher than 50% are printed at the branch nodes.

**Figure 2 pathogens-11-00979-f002:**
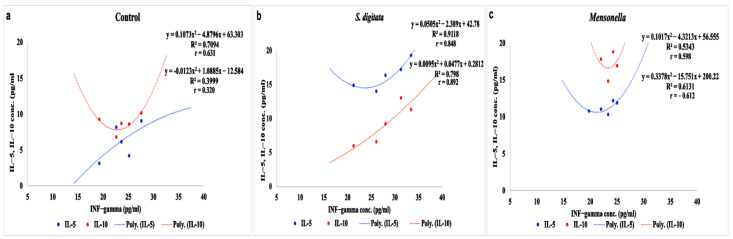
Correlation between INF-gamma serum level and IL-5 and IL-10 levels. The level of INF-gamma correlated with the levels of IL-5 and IL-10 in control (**a**) and infected animals (**b**,**c**). In the control group, a moderate positive correlation was observed between INF-gamma and IL-5; however, a weak correlation between INF-gamma and IL-10 serum levels was recorded. A strong positive correlation was recorded between INF-gamma and IL-5 and IL-10 serum levels in animals infected with *S. digitata*. In contrast, a moderate positive correlation between INF-gamma and IL-5 serum levels and a negative moderate correlation between INF-gamma and IL-10 serum levels were observed in *Mansonella* sp. (*T*)-infected groups.

**Figure 3 pathogens-11-00979-f003:**
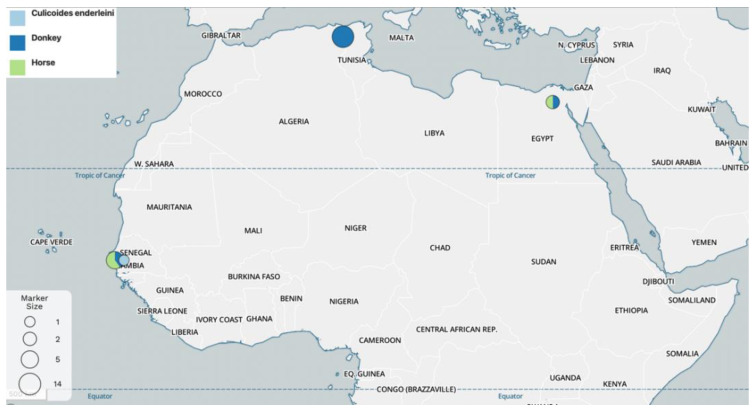
Geographical mapping of the available molecular data (i.e., 18S, ITS1, cox1 and 5S) on *Mansonella (T.)* sp., genotype OM-2015 based on host source. The map was generated using Microreact server (available at: https://microreact.org accessed on 25 May 2022). Maps © Mapbox (www.mapbox.com/about/maps).

**Figure 4 pathogens-11-00979-f004:**
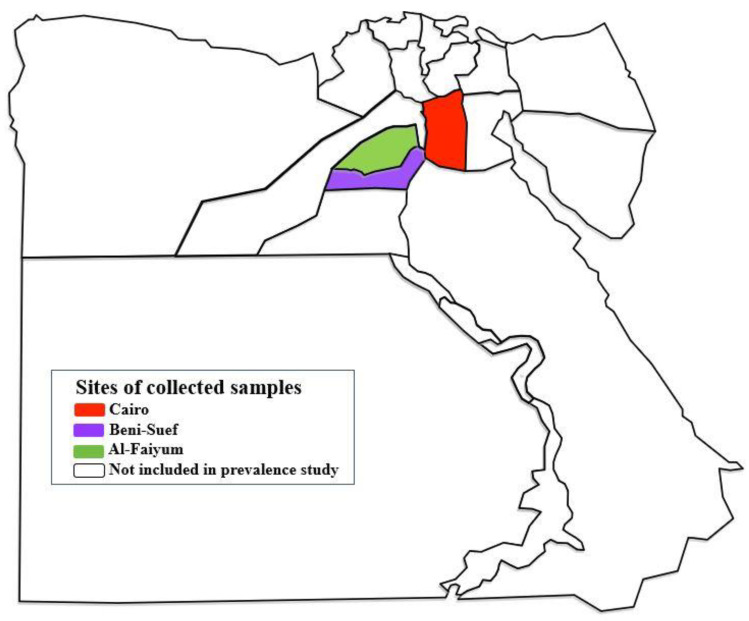
Map of Egypt shows the locations of provinces where *Equidae* blood samples were collected. https://en.wikipedia.org/wiki/Governorates_of_Egypt (accessed on 25 May 2022) and the picture has CC BY-SA 3.0.

**Table 1 pathogens-11-00979-t001:** Results of molecular identification of equine filaroids from the analysed samples (i.e., blood or filaroid worms). Species identification, prevalence and sequence accession number are shown.

Animal Hosts (no.)	No. of Filaroid-Positive Animals (%)	Species Name (Acc. No., Size bps)	100% Identical Sequences from GenBank
Horses (320)	1 (0.3)	*Mansonella**(T.)* sp. (MN728216)	*Mansonella* sp. (MT786947-49)
Total	1 (0.3)		
Donkeys (109)	9 (8.3)	*Mansonella**(T.)* sp. (MN728182, MN728181 *)	*Mansonella* sp. (MT786947-49)
	5 (4.6)	*Setaria digitata* (MN728184, MN728183 **)	*S. digitata* (DQ094175)
	1 (0.9)	*Dirofilaria repens* (MN728180, MN728180 *)	*D. repens* (MN728215)
Total	15 (13.8)		

* and ** indicate accession numbers of sequence of the filaroid worms from one infected donkey or two infected donkeys, respectively.

**Table 2 pathogens-11-00979-t002:** Serum cytokines profile of *Mansonella*
*(T)* sp.- and *S. digitata*-infected equids.

	INF-gamma (pg/mL)	IL-5 (pg/mL)	Th1/Th2 (INF-gamma/IL-5)	IL-10 (pg/mL)
Control	23.634 ± 1.66	6.160 ± 0.34	3.84	8.726 ± 2.11
*Mansonella**(T)* sp.	21.992 ± 3.18	11.054 ± 2.61 ^a^	1.99	17.868 ± 1.02 ^bB^
*S. digitata*	28.006 ± 1.09 ^aA^	16.362 ± 1.35 ^bA^	1.71	9.254 ± 2.99

a and b indicate significant difference in cytokines concentration at *p*-value of 0.01 and 0.05, respectively, between infected and control animals. A and B indicate significant difference in cytokine profile at *p*-value of 0.01 and 0.05, respectively, between animals of the same host species infected with different filaroid species (*Mansonella (T)* sp. and *S. digitata*).

**Table 3 pathogens-11-00979-t003:** The data of collected samples.

Provinces	Lat./Long.	Hosts	Locations	Total
Cairo	30°03′45.47″ N, 31°14′58.81″ E	Horses	Police Academy (El-Abbasia)	94
			Police Academy (El-Tagamoa)	70
			Police Academy (El-Pasateen)	147
Beni-Suef	29°03′60.00″ N, 31°04′60.00″ E	Horses	households	9
		Donkeys	households	22
Al-Faiyum	29°18′35.82″ N, 30°50′30.48″ E	Donkeys	households	87

## Data Availability

Not applicable.

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
