# Peer review of "Emergent and Neglected Equine Filariosis in Egypt: Species Diversity and Host Immune Response"

_pathogens, 2022, doi:10.3390/pathogens11090979_

Round 1

Reviewer 1 Report

In this manuscript the authors investigated equines in Egypt for the presence of filaroid nematode worms. They used molecular tools to detect and identify the parasites. They discovered worms of the genera Mansonella, Setaria and Dirofilaria and they found much higher prevalence of these worms in donkeys compared with horses. Finally, for each identified parasite species, the authors evaluated the immunological status of the infected host animals and compared it with uninfected animals and animals infected with a different filaroid worm. I am not an immunologist and can therefore not fully evaluate the technical validity of these experiments. The observed correlations are interesting but they should not be over interpreted. This could be discussed more clearly. In particular, the numbers of infected animals are relatively small and the authors do not provide any information if the animals infected with a particular worm might have been housed close together. If so, they might also have been exposed to a different common pathogen that might as well be the cause for the immunological state.

This article is straight forward but useful, however of more local and specialist interest. It is up to the editor to decide if it fits the scope of Pathogens or if it should be redirected to a more specialized veterinary journal (e.g. Veterinary Parasitology: Regional Studies and Reports). The manuscript is generally well-understandable and, as far as I can tell, the claims are supported by the data. Below, I listed a few, rather minor points, that should be addressed.

Specific points:

Line 82: remove "and" before "showed 100%"

Lines 85, 88: remove "and" before "were 100%"

Figure 1: I suggest printing the boot strap values into the figure. At least in my copy, the color coding is difficult to see. In particular, 70 and 80 as well as 90 and 100 are not distinguishable.

Lines 81 - 84, 151,152 and Figure 1: Could the authors comment on the taxonomy of S digitata and S. equina and why they identified their samples as S digitata and not S. equina? In the figure S. equina(PQVD01000119) looks equally close to the new samples as S digitata (DQ094175).

Line 118: Table 2 not Table 3

Lines 134,135: this sentence is unclear: I suggest something like: ...and 0.05, respectively, between individuals of the same host species infected with different filaroid pathogens ....).

Figure 2: For non-immunologists, like myself, please expand the legend and explain in more detail what is shown, in particular what the curves are.

Figure 2: As the font appears in my printed copy, it is too small.

Lines 153,154 and Table 3: The numbers in the text and in Table 3 do not match. I assume that the number of horses from El-Tagamoa is missing from the table. The text says that there were 320 horses from Cairo but the table lists that 9 of them were from Bei-Suef.

Line 182,183: Again, the numbers in the text and in the table do not match. The text claims that there were 109 donkeys from Al-Faiyum but the table lists only 87 donkeys from this province while the remaining 22 were from Beni-Suef.

Line 191,192: Again, the authors talk about one province but list the total number of animals (109) in the entire study.

Line 210,211: I do not think that, based on the presented data, it can be claimed that Mansonella infection "resulted" in a significant elevation of IL-10. It is a correlation but no causality should be claimed (in other places the authors are more careful not to imply causality from correlation). 

Author Response

Reviewer 1:

The observed correlations are interesting but they should not be over interpreted. This could be discussed more clearly. In particular, the numbers of infected animals are relatively small and the authors do not provide any information if the animals infected with a particular worm might have been housed close together. If so, they might also have been exposed to a different common pathogen that might as well be the cause for the immunological state.

Author’s response:

The requested clarification has been done (Line 241-242 and 269-276).

Specific points:

Line 82: remove "and" before "showed 100%"

Lines 85, 88: remove "and" before "were 100%"

Author’s response:

All requested corrections have been done.

Figure 1: I suggest printing the bootstrap values into the figure. At least in my copy, the color coding is difficult to see. In particular, 70 and 80, as well as 90 and 100, are not distinguishable.

Author’s response:

Thank you for pointing out this issue. We have now printed the bootstraps percent at each branch-node.

Lines 81 - 84, 151,152 and Figure 1: Could the authors comment on the taxonomy of S digitata and S. equina and why they identified their samples as S digitata and not S. equina? In the figure S. equina(PQVD01000119) looks equally close to the new samples as S digitata (DQ094175).

Author’s response:        

We agree with the reviewer comment on this phylogenetic issue. In fact, the blast analysis of the S. digitata amplified in the present study showed 100% identity with the available S. digitata from GenBank database. While, after checking, the sequence of S. equina (PQVD01000119) derived from a WGS project number (BioProject: PRJNA432023) of an unverified sample. We have now removed it now from the tree to avoid any ambiguity.

Line 118: Table 2 not Table 3

Author’s response:

The requested correction has been done.

Lines 134,135: this sentence is unclear: I suggest something like: ...and 0.05, respectively, between individuals of the same host species infected with different filaroid pathogens ....).

Author’s response:

The legend has been modified.

Figure 2: For non-immunologists, like myself, please expand the legend and explain in more detail what is shown, in particular, what the curves are.

Author’s response:

More details have been added in the legend which explained previously in the text (Line 142-148).

Figure 2: As the font appears in my printed copy, it is too small.

Author’s response:

The new picture has been added with more clarifications.

Lines 153,154 and Table 3: The numbers in the text and in Table 3 do not match. I assume that the number of horses from El-Tagamoa is missing from the table. The text says that there were 320 horses from Cairo but the table lists that 9 of them were from Bei-Suef.

Author’s response:

We thank the Reviewer for this valuable remark. We have added numbers of horses which were collected from El-Tagamoa (70). So, the total number of horses was 320.

Line 182,183: Again, the numbers in the text and in the table do not match. The text claims that there were 109 donkeys from Al-Faiyum but the table lists only 87 donkeys from this province while the remaining 22 were from Beni-Suef.

Author’s response:

Total number of donkeys is 109, and it is correct in both text and table 3. Setaria digitata positive donkeys were 5 donkeys which derived from Al-Faiyum province, while the remaining provinces were free from S. digitata. The sentence has been modified to be more clear. 

Line 191,192: Again, the authors talk about one province but list the total number of animals (109) in the entire study.

Author’s response:

Total number of donkeys is 109, and it is correct in both text and table 3. Dirofilaria repens positive donkey was derived from Al-Faiyum province, while the remaining provinces free from D. repens. The sentence has modified to be more clear. 

Line 210,211: I do not think that, based on the presented data, it can be claimed that Mansonella infection "resulted" in a significant elevation of IL-10. It is a correlation but no causality should be claimed (in other places the authors are more careful not to imply causality from correlation). 

Author’s response:

Thanks for this valuable remark. The sentence has been edited in order to meet the reviewer's comment (Line 251).

Reviewer 2 Report

Line 56 - line 66: Introduction is to introduce the background knowledge of the relevant research in order to elicit the feasibility, purpose and significance of the current research. Obviously, the authors need to seriously modify it.

Line 67- line 73: What’s the purpose of this study?

Line 242- line 243: Why chose these three Egyptian provinces? geographic factor or others?

Table 3 didn't contain the species, age, or gender of sampling animals. Do these factors affect the results of study?

Line 262: blood samples? You mean, serum or whole blood?

Line 264: )?   Please notice the language errors in the manuscript, double check seriously.

Author Response

Reviewer 2:

Line 56 - Line 66: Introduction is to introduce the background knowledge of the relevant research in order to elicit the feasibility, purpose and significance of the current research. Obviously, the authors need to seriously modify it.

Author’s response:

This paragraph has been modified in order to justify the reviewer's comment (Line 63-71).

Line 67- Line 73: What’s the purpose of this study?

Author’s response:

The aim of the study has been added (Line 86-88).

Line 242- Line 243: Why chose these three Egyptian provinces? geographic factor or others?

Author’s response:

This study just explored equine filarioid diversity and their immune response. Also, we mentioned in the materials and methods section that a convenience sampling strategy was used for finding out the prevalence of filarioid species in equines at the time of the study through a cross-sectional sampling from the equines’ population.

Table 3 didn't contain the species, age, or gender of sampling animals. Do these factors affect the results of study?

Author’s response:

That is true, we collected information on the animals studied, but our study has mainly focused on the screening and characterization of equine filarioid species and their host immune response. We also discussed the prevalence of the infection of each equine filarioid species.

Therefore, the analysis of epidemiological factors will be investigated in further studies.  

Line 262: blood samples? You mean, serum or whole blood?

Author’s response:

In the animal sampling section, we mentioned both types of samples were collected, EDTA-blood samples for molecular investigation and serum samples for immunological studies (Line 295-297).

To be specified, EDTA-blood has been added (Line 280).

Line 264: )?   Please notice the language errors in the manuscript, double check seriously.

Author’s response:

The whole manuscript has been edited for the English language.

Round 2

Reviewer 2 Report

None.